# Beta Human Papillomavirus 8E6 Attenuates Non-Homologous End Joining by Hindering DNA-PKcs Activity

**DOI:** 10.3390/cancers12092356

**Published:** 2020-08-20

**Authors:** Changkun Hu, Taylor Bugbee, Monica Gamez, Nicholas A. Wallace

**Affiliations:** 1Division of Biology, Kansas State University, Manhattan, KS 66506, USA; chu1@ksu.edu (C.H.); tbugbee@ksu.edu (T.B.); 2Bristol Medical School, Translational Health Sciences, University of Bristol, Bristol BS1 3NY, UK; mg14606@bristol.ac.uk

**Keywords:** human papillomavirus, HPV, DNA repair, double strand breaks, non-homologous end joining

## Abstract

Cutaneous viral infections occur in a background of near continual exposure to environmental genotoxins, like UV radiation in sunlight. Failure to repair damaged DNA is an established driver of tumorigenesis and substantial cellular resources are devoted to repairing DNA lesions. Beta-human papillomaviruses (β-HPVs) attenuate DNA repair signaling. However, their role in human disease is unclear. Some have proposed that β-HPV promotes tumorigenesis, while others suggest that β-HPV protects against skin cancer. Most of the molecular evidence that β-HPV impairs DNA repair has been gained via characterization of the E6 protein from β-HPV 8 (β-HPV 8E6). Moreover, β-HPV 8E6 hinders DNA repair by binding and destabilizing p300, a transcription factor for multiple DNA repair genes. By reducing p300 availability, β-HPV 8E6 attenuates a major double strand DNA break (DSB) repair pathway, homologous recombination. Here, β-HPV 8E6 impairs another DSB repair pathway, non-homologous end joining (NHEJ). Specifically, β-HPV 8E6 acts by attenuating DNA-dependent protein kinase (DNA-PK) activity, a critical NHEJ kinase. This includes DNA-PK activation and the downstream of steps in the pathway associated with DNA-PK activity. Notably, β-HPV 8E6 inhibits NHEJ through p300 dependent and independent means. Together, these data expand the known genome destabilizing capabilities of β-HPV 8E6.

## 1. Introduction

Human papillomavirus (HPV) is a small double-stranded DNA virus family that infects mucosal and cutaneous epithelia. Currently, about 400 types of HPV have been identified [1]. This family is classified into five genera (alpha, beta, gamma, mu, and nu), based on the sequence of the L1 capsid gene [2,3,4,5]. Of these genera, the alpha genus of HPV has been most thoroughly characterized because members of this genus cause cervical cancer, head and neck cancer, and genital warts [2,4,6]. Despite their widespread presence in the skin, the contribution of members of the beta genus of HPV (β-HPV) to human disease is unclear. Moreover, β-HPVs were first isolated from patients with a rare genetic disorder, epidermodysplasia verruciformis (EV) [5,7]. In these individuals and people receiving immunosuppressive drugs after organ transplants, β-HPVs appear to promote non-melanoma skin cancer (NMSC) [5,8]. An array of in vivo and in vitro studies also support the role of β-HPVs in promoting NMSC [9,10].

However, there are questions about the oncogenic potential of β-HPV in the general population. In most immunocompetent individuals, β-HPV infections are transient [11]. Furthermore, β-HPV genes are rarely expressed in tumor tissue [12,13]. This ruled out traditional methods of viral oncogenesis where the tumor becomes dependent on continued viral gene expression and led to the so-called “hit and run” hypothesis [13]. This hypothesis proposes that β-HPVs promote early stages of tumorigenesis by destabilizing the host genome, leading to mutations that could drive oncogenesis without continued viral gene expression [14]. Although feasible, the model is difficult to test. As a result, it remains unclear if/how frequently β-HPV infections contribute to NMSC. The “hit and run” model faced further challenges when a recent report suggested β-HPV infections protected against NMSC [15]. The widespread nature of these infections and their contentious role in tumorigenesis are strong motivating factors for ongoing research into the basic biology of the virus and its gene products.

Among β-HPV proteins, the E6 or β-HPV E6 is best characterized. This report focuses on the E6 from β-HPV 8 (β-HPV 8E6), and its ability to disrupt DNA repair [16,17,18,19]. Interestingly, β-HPV 8E6 exerts its influence in part by binding and destabilizing the cellular histone acetyltransferase, p300 [20]. Moreover, p300 is a transcription factor necessary for robust expression of key DNA repair proteins (ATM, ATR, BRCA1 and BRCA2) [16,17,18]. When β-HPV 8E6 is present, the reduced availability of these repair factors makes UV-induced DNA lesions more persistent [17,18]. The inability to resolve these lesions increases the frequency of replication fork collapse and the generation of UV-induced double stranded DNA breaks (DSBs) [18]. These breaks occur during S and G2 phases of the cell cycle, when homologous recombination (HR) is the principle mechanism of DSB repair [21,22,23,24,25]. Despite increasing the need for HR, β-HPV 8E6 impairs the pathway by decreasing BRCA1 and BRCA2 expression and foci formation [16]. 

When HR fails, non-homologous end joining (NHEJ) serves as a backup repair mechanism. NHEJ is not restricted to any portion of the cell cycle, but tends to occur when HR is not available (i.e., G1 and early S phases) [26,27,28,29,30]. It is an error-prone pathway that initiates with 53BP1 binding to the DSB [31,32]. This simultaneously promotes NHEJ, while restricting HR [32,33,34,35,36,37]. After 53BP1 binding, a heterodimer of Ku70 and Ku80 is recruited to the damaged site, tethering at the exposed DNA ends [27]. Next, DNA-dependent protein kinase catalytic subunit (DNA-PKcs) binds to the Ku dimer to form a holoenzyme, known as DNA-PK. Then, DNA-PKcs becomes activated by auto-phosphorylation (pDNA-PKcs) [38,39,40]. Once activated, pDNA-PKcs facilitates NHEJ by phosphorylating/activating downstream targets, including Artemis, XRCC4, and DNA ligase IV (LIG4) [40,41,42,43,44]. Artemis has both exonuclease and endonuclease activity that processes DNA single-strand overhangs into blunt end NHEJ-ready substrates [45,46,47]. When overhanging DNA ends have been removed, the XRCC4-XLF-Ligase IV complex links the two DNA ends together [48,49].

Since limitations in HR are addressed with increases in NHEJ, it was reasonable to hypothesize that β-HPV 8E6 increased repair by NHEJ. However, this report presents contrary evidence to this idea. Furthermore, β-HPV 8E6 reduced NHEJ repair at a defined genomic location and reduced DNA-PKcs autophosphorylation. This culminated in more persistent DNA-PKcs foci and diminished pDNA-PKcs-dependent signaling events (phosphorylation of Artemis and XRCC4 repair complex formation). Interestingly, β-HPV 8E6 appears to diminish NHEJ activity, through both p300-dependent and -independent mechanisms.

## 2. Results

### 2.1. β-HPV 8E6 Decreases NHEJ Efficiency

We have previously shown that β-HPV 8E6 disrupts HR by destabilizing p300, a transcription factor for two HR genes (BRCA1 and BRCA2) [16]. NHEJ competes with HR for access to DSBs [26,50,51], suggesting that NHEJ may occur more frequently in cells expressing β-HPV 8E6. To test this, we measured NHEJ efficiency, using a recently described end-joining assay that uses CD4 expression as a readout [28]. In this assay, CAS9 endonucleases are used to create breaks in the human genome downstream of the GAPDH promoter, and upstream of the CD4 exon. These genes are oriented in the same direction and sit ~0.25 Mb apart. When NHEJ repairs the breaks, it results in a recombination event where CD4 expression is driven by the GAPDH promoter (Figure 1A). The CD4 promoter is typically inactive in cells outside of the immune system, which provides a low background in many cell types [52]. This assay was verified by measuring NHEJ efficiency in U2OS cells. CD4 expression was detected by immunoblot and normalized to the abundance of FLAG-tagged CAS9, as a control for transfection efficiency (Figure 1B,C). As previously reported, expression of CAS9 endonucleases targeting GAPDH and CD4 lead to CD4 expression. The assay was further verified by treating U2OS cells with 10 μM ATM inhibitor (KU55933) and 10 μM DNA-PKcs inhibitor (NU7441), that are known to increase and decrease NHEJ, respectively [28,38]. As expected, KU55933 increased CD4 expression, while NU7441 decreased it (Figure 1B,C). Next, the impact of β-HPV 8E6 on NHEJ was determined in previously described U2OS cells, expressing β-HPV 8E6 (U2OS β-HPV 8E6) or vector control (U2OS LXSN). Unexpectedly, β-HPV 8E6 significantly decreased CD4 compared to U2OS LXSN cells (Figure 1D,E). The p300-dependence of this phenotype was also probed in U2OS cells, expressing a mutant β-HPV 8E6 (β-HPV Δ8E6), where 5 amino acids responsible for p300 binding (Residues 132–136) were deleted [20]. NHEJ frequency was also decreased in these cells (U2OS β-HPV Δ8E6) (Figure 1D,E). Neither inhibitor nor expression of wild type or mutant β-HPV 8E6 significantly altered transfection efficiency (Appendix A). To determine if these results were reproducible in a more physiological relevant cell line, the assay was repeated in a pair of previously described telomerase immortalized human foreskin keratinocytes (HFK) cell lines [53]. Again, β-HPV 8E6 expressing HFK cells (HFK β-HPV 8E6) had reduced NHEJ efficiency, compared with vector control HFK cells (HFK LXSN, Figure 1F,G). Together, these data indicate that β-HPV 8E6 hinders NHEJ, through p300-independent mechanisms. However, they cannot rule out the possibility that β-HPV 8E6 also acts through a p300-dependent mechanism.

### 2.2. β-HPV 8E6 Attenuates DNA-PKcs Phosphorylation

β-HPV 8E6 prevents the repair of UV lesions and completion of HR by reducing the abundance of key repair factors (ATM, ATR, BRCA1 and BRCA2) [16,17,18]. This suggests that β-HPV 8E6 may act through a similar mechanism to impair NHEJ. To assess this possibility, the abundance of canonical NHEJ proteins was determined in HFK. In untreated HFK cells, β-HPV 8E6 did not decrease Ku80, DNA-PKcs, Artemis, XRCC4, or Ligase IV abundance (Appendix A). These data suggest that β-HPV 8E6 exerted its influence in a post-translational manner, so DNA-PKcs activation (via autophosphorylation at S2056) was assessed in cells exposed to Zeocin, a radiomimetic [54,55]. This modification was chosen because phosphorylated DNA-PKcs or pDNA-PKcs is a well characterized and early step in NHEJ [39,56]. Cells were treated with designated Zeocin concentrations for 24 hours (Figure 2). pDNA-PKcs increased in a Zeocin dose-dependent manner in HFK LXSN. However, this response was attenuated in HFK β-HPV 8E6 (Figure 2A,B, Appendix A). Similar results were observed when the experiment was repeated in U2OS LXSN and U2OS β-HPV 8E6 cells. However, U2OS expressing β-HPV Δ8E6 (U2OS β-HPV Δ8E6) behaved like U2OS LXSN cells, by increasing the proportion of activated DNA-PKcs in response to Zeocin exposure (Figure 2C,D). These data suggest that β-HPV 8E6 impairs DNA-PKcs activation in a p300-dependent manner. pDNA-PKcs and total DNA-PKcs were separately normalized to GAPDH (Appendix A). Consistently, β-HPV 8E6 attenuated DNA-PKcs phosphorylation, in both HFK and U2OS. Zeocin exposure consistently decreased total DNA-PKcs. However, β-HPV 8E6 did not statistically significantly change this decrease (Appendix A).

To probe the breadth of DNA-PKcs inhibition, another genotoxic reagent (hydrogen peroxide, or H_2_O_2_) was used to activate NHEJ in U2OS LXSN, U2OS β-HPV 8E6 and U2OS β-HPV Δ8E6 cells. Unlike Zeocin, which induces breaks by intercalating into base pairs and causing cleavage, H_2_O_2_ generates DSBs by generating reactive oxygen species (ROS) [54,55]. This represents a more physiological type of DSB, as ROS are caused by cell metabolism [57]. β-HPV 8E6 and β-HPV Δ8E6 blunted pDNA-PKcs in response to H_2_O_2_ (Appendix A).

To facilitate repair, pDNA-PKcs must localize to a DSBs. This localization results in complexes that are detectable as foci by immunofluorescence (IF) microscopy. These foci are indicative of ongoing repair. pDNA-PKcs foci were readily detected in untreated HFK LXSN cells, but less frequent in HFK β-HPV 8E6 cells (Figure 3A,B). Prior reports found that β-HPV 8E6 increased the frequency of DSBs in untreated cells, suggesting that the reduced pDNA-PKcs foci are unlikely to indicate genomic stabilization [16]. An alternative explanation consistent with the data shown in Figure 1 is that β-HPV 8E6 reduced the frequency of NHEJ. When repair complexes are not resolved, the repair proteins spread along nearby chromatin producing larger/brighter foci [58,59,60]. As a result, foci intensity was used as an indicator of repair efficiency (brighter foci indicate more persistent lesions). Consistent with NHEJ inhibition, β-HPV E6 increased pDNA-PKcs foci intensity in HFKs (Figure 3A,C). Similar results were obtained in U2OS cells (Figure 3D–F). Interestingly, β-HPV Δ8E6 did not alter pDNA-PKcs foci prevalence or intensity (Figure 3D–F). Together, our data suggest that β-HPV 8E6 hinders DNA-PKcs activation in a p300-dependent manner (Figure 1, Figure 2 and Figure 3), but also can impair NHEJ through a p300-independent mechanism(s) (Figure 1D,E). 

### 2.3. β-HPV 8E6 Attenuates DNA-PKcs-Dependent Signaling

To further determine the ability of β-HPV 8E6 to alter pDNA-PKcs signaling, DSBs were induced with Zeocin (10 μg/mL), then observed with immunofluorescence microscopy. pDNA-PKcs foci appeared rapidly in HFK LXSN cells and reached their maxima approximately one hour after Zeocin exposure (Figure 4A,B). Twenty-four hours later, the pDNA-PKcs foci had returned to background levels. β-HPV 8E6 did not alter the initial induction of pDNA-PKcs foci by Zeocin. However, pDNA-PKcs foci were significantly more persistent in HFK β-HPV 8E6 cells. Similar results were obtained in U2OS LXSN and U2OS β-HPV 8E6 cells (Figure 4C,D). Consistent with a p300-dependent mechanism, pDNA-PKcs foci kinetics were similar in U2OS β-HPV Δ8E6 and U2OS LXSN cells after Zeocin exposure (100 μg/mL). Supporting the idea that pDNA-PKcs foci represent active repair complexes, while pDNA-PKcs was detected in damage induced foci, total DNA-PKcs showed pan-nuclear staining in treated and untreated cells (Appendix A). 

Having seen β-HPV 8E6 impair DNA-PKcs autophosphorylation and repair complex resolution, the ability of β-HPV 8E6 to hinder other DNA-PKcs-dependent steps in NHEJ was determined. Published reports indicated that Artemis is a DSB-induced target of DNA-PKcs phosphorylation at Serine 516 (pArtemis) [46,47]. This relationship was confirmed using immunoblots to detect pArtemis when DNA-PKcs activity was blocked with a small molecule inhibitor (1 μM NU7441). While pArtemis levels rose in a Zeocin dose-dependent manner in wild type cells, pArtemis abundance was limited by the inhibitor (Appendix A). Having confirmed that Artemis phosphorylation depended on DNA-PKcs activity, the extent that β-HPV 8E6 reduced phosphorylation of Artemis in response to Zeocin was defined. β-HPV 8E6 blocked Artemis phosphorylation in HFKs (Figure 5A,B, Appendix A). These results were also reproducible in U2OS (Figure 5C,D, Appendix A). Notably, pArtemis levels rose in U2OS LXSN and U2OS β-HPV △8E6 cells in response to Zeocin. These data indicate that β-HPV 8E6 ’s p300-dependent attenuation of DNA-PKcs-dependent signaling extended to Artemis activation. 

To better understand the extent that NHEJ was impaired by β-HPV 8E6 hindered NHEJ, the ability of XRCC4 to localize to sites of damage was assessed. This occurs downstream of Artemis activation and is required for the DNA ligation step in NHEJ [49,61,62]. Like Artemis, XRCC4 is also a substrate of DNA-PKcs [27,28]. However, the role of that phosphorylation is poorly understood [63,64]. A study showed that DNA ligation fails without XRCC4, because it is required for LIG4 stabilization [65]. In HFK LXSN cells, Zeocin induced XRCC4 foci (detected by IF microscopy) and were readily resolved (Figure 5E,F). However, β-HPV 8E6 prevented an induction of XRCC4 foci in response to Zeocin. These results were repeated in U2OS cells (Figure 5G,H). Interestingly, U2OS β-HPV Δ8E6 also decreased XRCC4 recruitment, which may partially explain the p300-independent mechanism that β-HPV 8E6 diminishes NHEJ efficiency. Together, these data suggest that β-HPV 8E6 impairs XRCC4 recruitment to sites of damage.

### 2.4. p300 is Required for Robust DNA-PKcs Signaling

The data above suggest that p300 is required for DNA-PKcs-dependent NHEJ. To confirm this relationship, NHEJ and DNA-PKcs signaling was assessed in previously described p300 competent (p300 WT) and p300 knockout (p300 KO) HCT116 cells [66]. p300 KO HCT116 cells were notably less capable of initiating and completing the pathway. Specifically, the CD4 reporter assay (described in Figure 1A) found reduced NHEJ in p300 KO HCT116 cells (Figure 6A,B). While p300 knockout did not change basal DNA-PKcs phosphorylation (Appendix A), immunoblots indicate that it hindered DNA-PKcs activation (pDNA-PKcs), following Zeocin exposure (Figure 6C,D, Appendix A). Loss of p300 also increased pDNA-PKcs foci persistence (Figure 6E,F). Finally, p300 knockout attenuated Artemis phosphorylation in response to DSB induction (Figure 6G,H, Appendix A). These data demonstrate p300′s requirement in NHEJ and DNA-PKcs-dependent signaling.

## 3. Discussion

Because β-HPV 8E6 attenuated the repair of DSBs by HR [16], we initially hypothesized that this would make cells more likely to use the NHEJ pathway. NHEJ is prone to mutations, because it requires blunt ends as a substrate for repair. Typically, when NHEJ initiates, a Ku70/Ku80/DNA-PKcs trimer localizes to the lesion (Figure 7A). Once becoming activated via autophosphorylation, DNA-PKcs then promotes the pathway’s progression via phosphorylating downstream repair components. The phosphorylation of Artemis leads to resection of any overhanging DNA. Finally, XRCC4, XLF and LIG4 form a trimer at the newly blunted ends and ligate them together, fixing the break [48,49,67]. This was not the case in cells expressing β-HPV 8E6 (Figure 7B). β-HPV 8E6 reduced DNA-PKcs autophosphorylation (Figure 2) and increased the persistence of DNA-PKcs localized to DNA damage (Figure 3 and Figure 4). In turn, DNA-PKcs’ phosphorylation of Artemis was reduced and XRCC4 was less able to form repair complexes in response to DSB-induction (Figure 5). A reporter assay confirmed that these defects resulted in a reduced ability to repair DSBs via NHEJ.

β-HPV 8E6 hinders NHEJ, at least in part, by binding and destabilizing p300. p300 functions as a transcription factor for repair gene expression [68,69]. By reducing p300 availability, β-HPV 8E6 lowers the abundance of at least four DNA repair factors (BRCA1, BRCA2, ATR, and ATM) [16,17,18]. This manifests in a limited ability to respond to UV damage, or to utilize the HR pathway. In contrast, p300 does not appear to be a transcription factor for canonical NHEJ genes (Appendix A). Nevertheless, p300 is clearly required for robust NHEJ (Figure 6). Specifically, p300 promotes DNA-PKcs activity. Although the specific mechanistic explanation for our observations are not fully resolved, a prior study showed that p300 is required for the recruitment of Ku70/80 [70]. This may explain our observations, as Ku70/80 form a holoenzyme with DNA-PKcs to facilitate DNA-PKcs-mediated phosphorylation. However, our data rule out the possibility that p300 is needed for DNA-PKcs to localize to sites of damage. Instead, in the absence of p300, DNA-PKcs repair complexes become more persistent. DNA-PKcs activity requires acetylation, but the histone acetyltransferase was not determined [71]. Perhaps p300 is responsible for the post-translational modification. Setting aside these unknowns, our data demonstrate that p300 is required for the completion rather than initiation of NHEJ. Interestingly, the NHEJ reporter assay indicates that β-HPV 8E6 also impairs NHEJ independently of p300 binding or reduced DNA-PKcs activity (Figure 1). Our data suggest a possible mechanism. β-HPV Δ8E6 retains the ability to hinder XRCC4 foci formation (Figure 5G,H) which would limit NHEJ independent of p300 destabilization. Furthermore, these data provide confirmation that the β-HPV Δ8E6 mutant retains some functionality.

The evidence provided here shows that β-HPV 8E6 diminishes essential NHEJ events, including DNA-PKcs phosphorylation at S2056. However, our efforts fall well short of resolving the role of β-HPV in NMSC development. Granted, the reduced DNA repair potential associated with β-HPV 8E6 would not be desirable in cutaneous tissue, as our skin protects against external mutagens. Supporting this assertion, previous studies have shown that pharmacological inhibition of DNA-PKcs increases mutagenesis [72]. Furthermore, DNA-PKcs inhibitors and DNA-PKcs inactivating mutations sensitize in vitro and animal models to radiation [43,73,74]. Given the importance of DNA-PKcs in protecting genome fidelity, β-HPV infections could increase mutations in skin cells after UV exposure. However, given the typically transient nature of β-HPV infections, the increased mutational burden may not be particularly consequential. Furthermore, others have suggested that β-HPV infections prime the immune system, helping to prevent NMSCs [15]. These positions are not mutually exclusive, and should not be interpreted as being in conflict. Perhaps the oncogenic consequences of β-HPV associated repair inhibition are limited to specific circumstances (e.g., immune suppression). One other difference in the two studies is that Strickley and colleagues used a mouse papillomavirus that does not bind p300 [75].

Accumulating evidence shows that β-HPV E6 increases the mutagenic potential of UV. This includes increasing the frequency with which UV causes DSBs and hindering repair of these deleterious lesions. Both error-free HR and error-prone NHEJ are impaired when β-HPV 8E6 is expressed. However, β-HPV 8E6 does not appear to limit their initiation, as evidenced by the formation of both RAD51 [16] and pDNA-PKcs repair complexes (Figure 4). If the initiation of NHEJ and HR were to occur at the same DSB, it would be problematic, as the two pathways are intrinsically incompatible. HR begins by generating a large single-stranded DNA overhang, while NHEJ starts by removing any overhangs. This could result in large deletions, as repair osculates between the two DSB repair pathways. Furthermore, despite attenuated HR and NHEJ, β-HPV 8E6 expressing cells eventually resolve most DSBs. This suggests that β-HPV 8E6 could force DSB repair, to occur by a less efficient and/or more mutagenic pathway. Our future directions include defining the dominant mechanisms of DSB repair in cells expressing β-HPV 8E6, and determining the mutagenic consequences of β-HPV 8E6 on DSB repair.

## 4. Materials and Methods 

### 4.1. Cell Culture and Reagents

Immortalized human foreskin keratinocytes (HFK), provided by Michael Underbrink (University of Texas Medical Branch, Galveston, TX, USA), were grown in EpiLife medium (Gibco, Gaithersburg, MD, USA), supplemented with 60 µM calcium chloride (Gibco), human keratinocyte growth supplement (Gibco), and 1% penicillin-streptomycin (Caisson, Smithfield, UT, USA). U2OS and HCT116 cells were maintained in DMEM supplemented with 10% FBS and 1% penicillin-streptomycin. Zeocin (Alfa Aesar, Ward Hill, MA, USA) and H2O2 were used to induce DSBs. NU7441 (Selleckchem) was used to inhibit DNA-PKcs phosphorylation. KU55933 (Selleckchem, Houston, TX, USA) was used to inhibit ATM kinase activity.

### 4.2. Immunoblotting

After being washed with ice-cold PBS, cells were lysed with RIPA Lysis Buffer (VWR Life Science, Philadelphia, PA, USA), supplemented with Phosphatase Inhibitor Cocktail 2 (Sigma, St. Louis, MO, USA) and Protease Inhibitor Cocktail (Bimake, Houston, TX, USA). The Pierce BCA Protein Assay Kit (Thermo Scientific, Waltham, MA, USA) was used to determine protein concentration. Equal protein lysates were run on Novex 3–8% Tris-acetate 15 Well Mini Gels (Invitrogen, Carlsbad, CA, USA) and transferred to Immobilon-P membranes (Millipore, Burlington, MA, USA). Membranes were then probed with the following primary antibodies: GAPDH (Santa Cruz Biotechnologies, Dallas, TX, USA), DNA-PKcs (abcam, Cambridge, UK), phospho DNA-PKcs S2056 (abcam), Artemis (abcam), phospho Artemis S516 (abcam), XRCC4 (Santa Cruz Biotechnologies), Ligase IV (abcam), CD4 (abcam), and DYKDDDDK (FLAG) Tag (Invitrogen). After exposure to the matching HRP-conjugated secondary antibody, cells were visualized using SuperSignal West Femto Maximum Sensitivity Substrate (Thermo Scientific).

### 4.3. Immunofluorescence Microscopy

Cells were seeded onto either 96-well glass-bottom plates (Cellvis) or coverslips, and grown overnight. Cells treated with Zeocin for specified time and concentration were fixed with 4% formaldehyde. Then, 0.1% Triton-X solution in PBS was used to permeabilize the cells, followed by blocking with 3% bovine serum albumin in PBS for 30 minutes. Cells were then incubated with the following antibodies: phospho DNA-PKcs S2056 (abcam), XRCC4 (Santa Cruz Biotechnology). The cells were washed and stained with the appropriate secondary antibodies: Alexa Fluor 594 goat anti-rabbit (Thermo Scientific A11012), Alexa Fluor 488 goat anti-mouse (Thermo Scientific A11001). After washing, the cells were stained with 30 µM DAPI in PBS, and visualized with the Zeiss LSM 770 microscope. Images were analyzed using the ImageJ techniques previously described [3].

### 4.4. End Joining Reporter Assay

The reporter assay used a previously described protocol [38], with the following modifications. Cells were seeded into 6-well plates. After transfection, CD4 expression was measured by immunoblotting.

### 4.5. Statistical Analysis

All values are represented as mean ± standard error (SE) from at least three independent experiments. Statistical differences between groups were measured by using Student’s *t*-test. *p*-values in all experiments were considered significant, at less than 0.05.

## 5. Conclusions

Accumulating evidence shows that β-HPV 8E6 reduces genome stability by disrupting DNA damage response. Particularly, β-HPV 8E6 disrupts homologous recombination, which is a major DSB repair pathway in the S phase and G2 phase of the cell cycle. The data presented here show that β-HPV 8E6 diminishes NHEJ, which can occur throughout the cell cycle. This expands β-HPV 8E6′s influence over DSB repair throughout the cell cycle. Finally, this work demonstrates that β-HPV 8E6 uses p300-dependent and p300-independent mechanisms to disrupt DNA repair. This suggests that there are considerable evolutionary forces driving β-HPV to hinder cellular responses to damaged DNA.

## Figures and Tables

**Figure 1 cancers-12-02356-f001:**
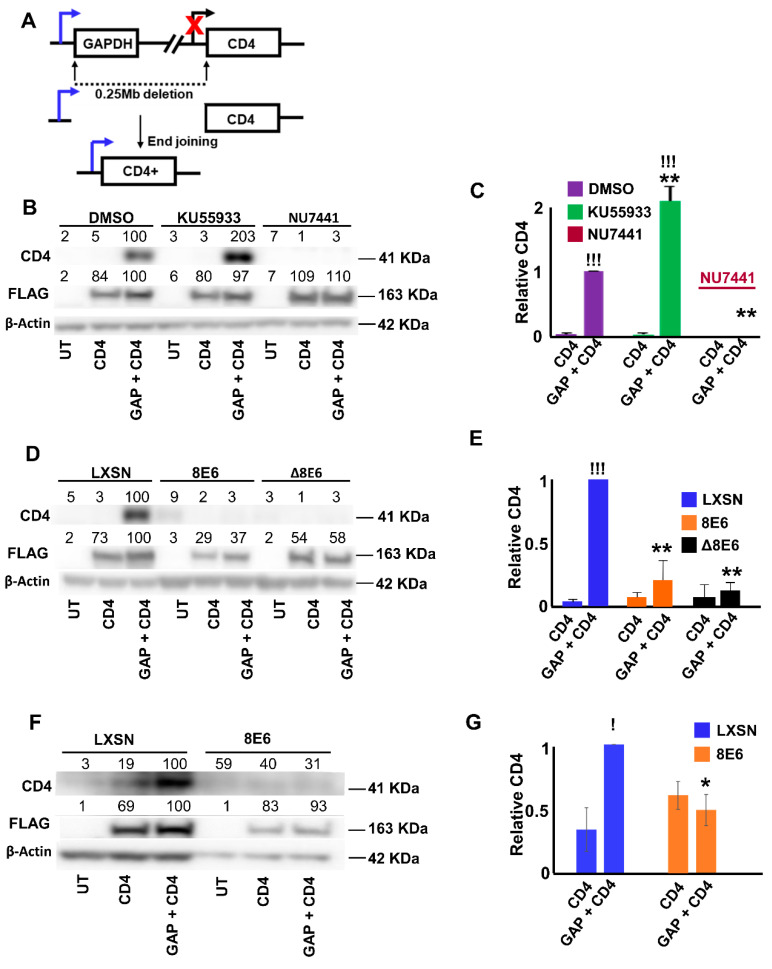
Moreover, beta-human papillomavirus (β-HPV) 8E6 decreases non-homologous end joining (NHEJ) efficiency, using CD4 expression as a readout. (**A**) Schematic of end-joining reporter assay. FLAG-tagged SgRNA-CAS9 induced double strand breaks in GAPDH and CD4 on chromosome 12 in U2OS cells. Rearrangement leads to CD4 expression driven by the promoter of GAPDH. Red “X” represents that CD4 expression is naturally inactivated. (**B**) Representative immunoblots showing CD4 expression in U2OS cells treated with control, ATM inhibitor (KU55933), and DNA-PK inhibitor (NU7441), after transfection with control (UT), FLAG-tagged SgRNA-CAS9 targeting CD4 (CD4), and FLAG-tagged SgRNA-CAS9 targeting GAPDH together with FLAG-tagged SgRNA-CAS9 targeting CD4 (GAP/CD4). (**C**) Densitometry of immunoblots (*n* = 3) from panel B. CD4 was normalized to β-actin as a loading control. Transfection efficiency was accounted for using FLAG abundance. (**D**) Representative immunoblots showing CD4 expression in U2OS LXSN, β-HPV 8 E6, and β-HPV Δ8E6 after transfection with control (UT), FLAG-tagged SgRNA-CAS9 targeting CD4 (CD4), and FLAG-tagged SgRNA-CAS9 targeting GAPDH and FLAG-tagged SgRNA-CAS9 targeting CD4 (GAP/CD4). (**E**) Densitometry of immunoblots (*n* = 3) from panel D. CD4 was normalized to β-actin as a loading control. Transfection efficiency was accounted for using FLAG abundance. (**F**) Representative immunoblots showing CD4 expression in HFK LXSN and HFK β-HPV 8 E6, after transfection with control (UT), FLAG-tagged SgRNA-CAS9 targeting CD4 (CD4), and FLAG-tagged SgRNA-CAS9 targeting GAPDH and FLAG-tagged SgRNA-CAS9 targeting CD4 (GAP/CD4). (**G**) Densitometry of immunoblots (*n* = 3) from panel F. CD4 was normalized to β-actin as a loading control. Transfection efficiency was accounted for using FLAG abundance. All values are represented as mean ± standard error from at least three independent experiments. Statistical differences between groups were measured by using Student’s *t*-test. * indicates *p* < 0.05. ** indicates *p* < 0.01. ! indicates significant difference between transfection with SgRNA-CAS9 targeting CD4 and co-transfection with SgRNA-CAS9 targeting CD4 and GAPDH. !!! indicates significant difference between transfection with SgRNA-CAS9 targeting CD4 and co-transfection with SgRNA-CAS9 targeting CD4 and GAPDH (*p* < 0.001).

**Figure 2 cancers-12-02356-f002:**
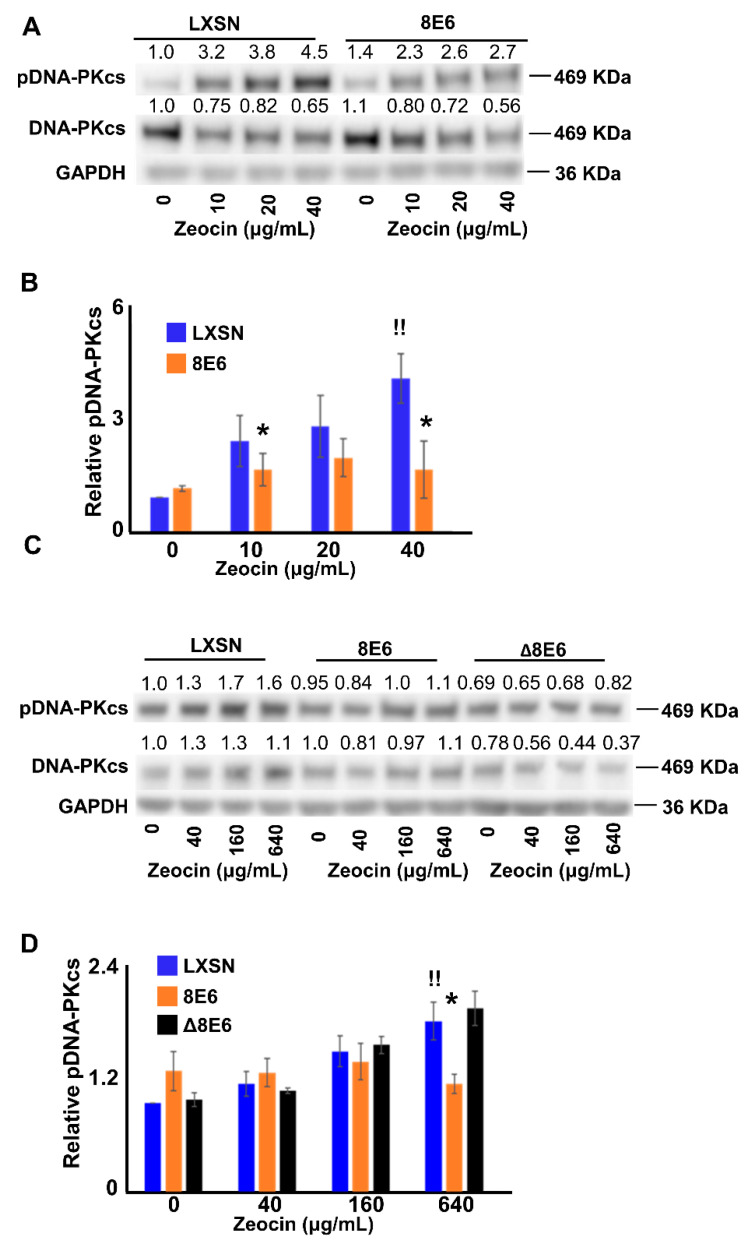
β-HPV 8E6 attenuates DNA-PKcs phosphorylation. (**A**) Representative immunoblot showing phospho-DNA-PKcs (pDNA-PKcs) and total DNA-PKcs in HFK LXSN and HFK β-HPV 8 E6. (**B**) Densitometry of immunoblots of pDNA-PKcs normalized to total DNA-PKcs and GAPDH as a loading control. (**C**) Representative immunoblot showing pDNA-PKcs and DNA-PKcs in U2OS LXSN, U2OS β-HPV 8 E6, and U2OS β-HPV Δ8 E6. (**D**) Densitometry of immunoblots (*n* = 4) of pDNA-PKcs normalized to total DNA-PKcs and GAPDH as a loading control. All values are represented as mean ± standard error from at least three independent experiments. Statistical differences between groups were measured by using Student’s *t*-test. * indicates *p* < 0.05. !! indicates significant difference between Zeocin treated and untreated group.

**Figure 3 cancers-12-02356-f003:**
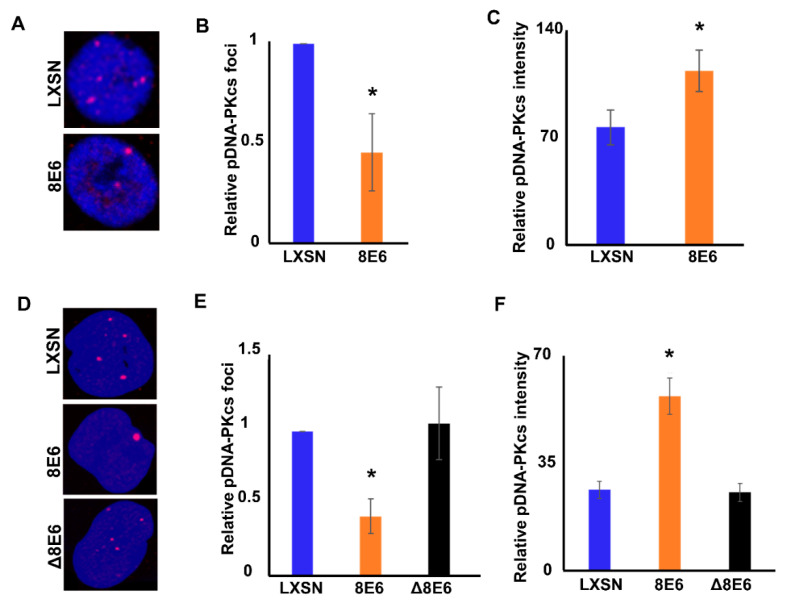
β-HPV 8E6 increases pDNA-PKcs foci size, but decreases frequency in untreated cells. (**A**) Representative images of pDNA-PKcs foci in HFK LXSN and HFK β-HPV 8 E6. (**B**) Percentages of cells with one or more pDNA-PKcs foci. (**C**) Average pDNA-PKcs foci intensity of HFK cells. (**D**) Representative images of pDNA-PKcs foci in U2OS LXSN, U2OS β-HPV 8 E6, and U2OS β-HPV Δ8 E6. (**E**) Percentages of cells with one or more pDNA-PKcs foci. (**F**) Average pDNA-PKcs focus intensity in U2OS cells. All values are represented as mean ± standard error from at least three independent experiments. Statistical differences between groups were measured by using Student’s *t*-test. * indicates *p* < 0.05. All microscopy images are 400X magnification.

**Figure 4 cancers-12-02356-f004:**
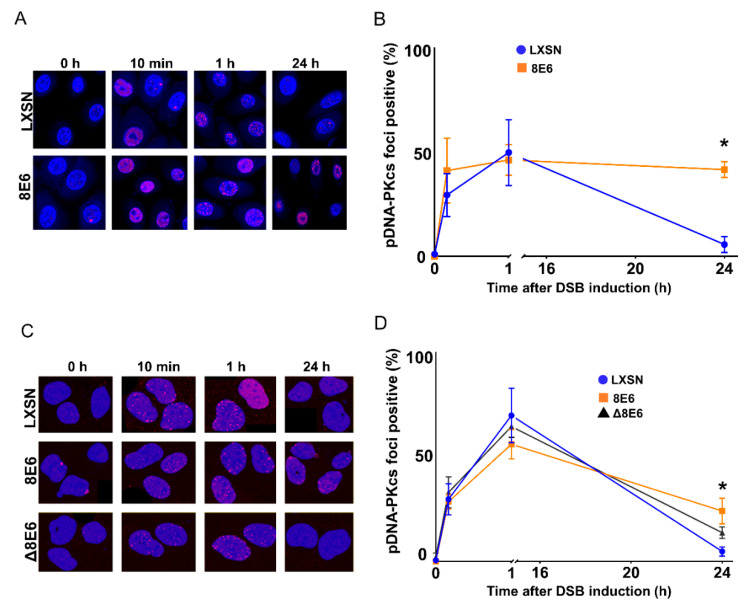
β-HPV 8E6 increases pDNA-PKcs foci persistence. Immunofluorescence microscopy was used to detect pDNA-PKcs foci in cells that were treated with Zeocin. (**A**) Representative images of pDNA-PKcs foci in HFK cell lines, following treatment with Zeocin for 10 min. (**B**) Percentage of pDNA-PKcs foci positive (> 2) cells following Zeocin exposure. (**C**) Representative images of pDNA-PKcs foci in U2OS cell lines treatment with Zeocin for 1 h, then harvested 0 h, 10 min, 1 h, and 24 h after Zeocin treatment. (**D**) Percentage of pDNA-PKcs foci positive (>4) U2OS cells following Zeocin exposure. All values are represented as mean ± standard error from at least three independent experiments. Statistical differences between groups were measured by using Student’s *t*-test. * indicates *p* < 0.05. All microscopy images are 400X magnification.

**Figure 5 cancers-12-02356-f005:**
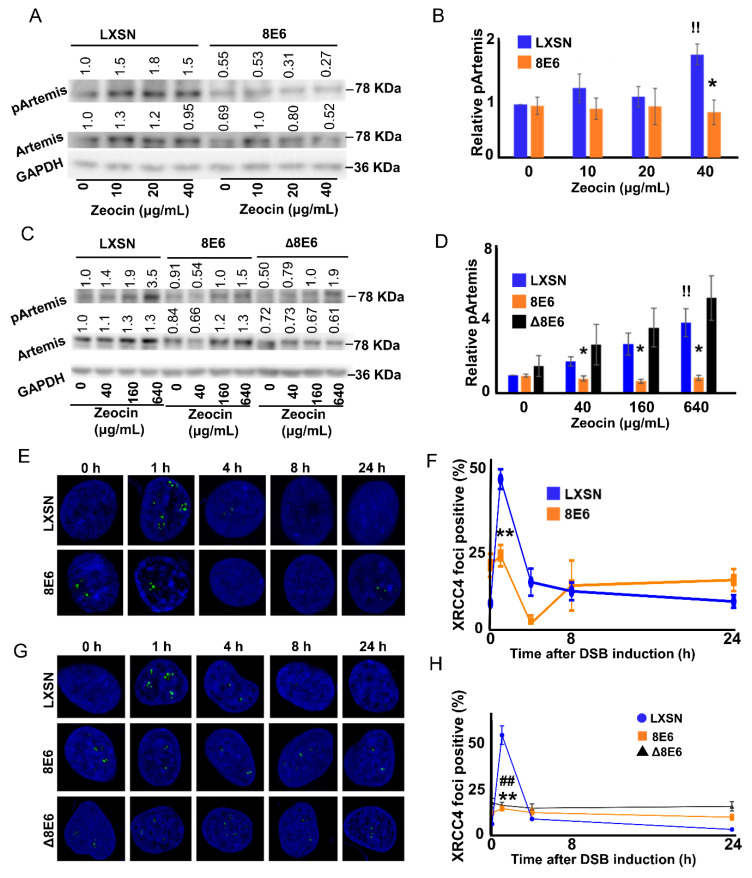
β-HPV 8E6 decreases pDNA-PKcs target proteins pArtemis abundance and XRCC4 foci formation. (**A**) Representative immunoblot showing pArtemis and total Artemis in HFK LXSN and HFK β-HPV 8 E6. (**B**) Densitometry of immunoblots (*n* = 4) of pArtemis normalized to total Artemis and to GAPDH as a loading control. (**C**) Representative immunoblot showing that pArtemis and total Artemis in U2OS LXSN, U2OS β-HPV 8 E6, or U2OS β-HPV Δ8 E6. (**D**) Densitometry of immunoblots (*n* = 4) of pArtemis normalized to total Artemis and to GAPDH as a loading control. (**E**) Representative images of XRCC4 foci in HFK cell lines 0–24 h following Zeocin exposure. (**F**) Percentages of XRCC4 foci positive (>2) HFK cells following DSB induction. (**G**) Representative images of XRCC4 foci in U2OS cell lines, 0–24 h following Zeocin exposure. All microscopy images are 400X magnification. (**H**) Percentages of XRCC4 foci positive (> 2) U2OS cells following DSB induction. All values are represented as mean ± standard error from at least three independent experiments. Statistical differences between groups were measured by using Student’s *t*-test. * indicates *p* < 0.05. ** indicates *p* < 0.01. !! indicates significant difference between Zeocin treated and untreated group. ## indicates significant difference between U2OS β-HPV Δ8 E6 and control.

**Figure 6 cancers-12-02356-f006:**
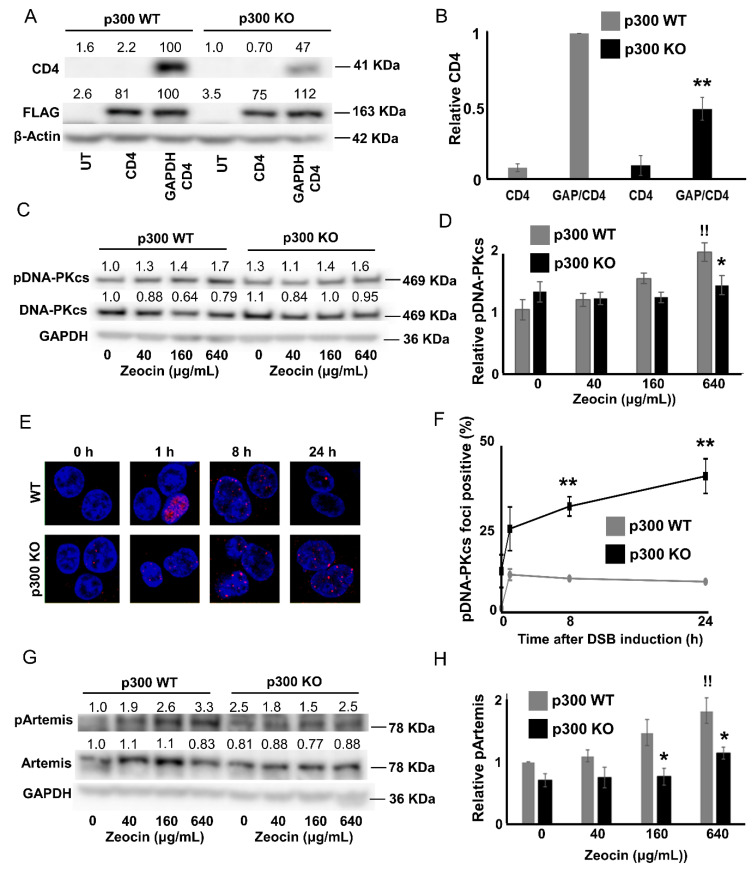
HCT116 P300 knockout decreases NHEJ efficiency. (**A**) Representative immunoblots showing CD4 expression in HCT116 p300 WT and HCT116 p300 KO after transfection with control (UT), FLAG-tagged SgRNA-CAS9 targeting CD4 (CD4), and FLAG-tagged SgRNA-CAS9 targeting GAPDH and FLAG-tagged SgRNA-CAS9 targeting CD4 (GAP/CD4). (**B**) Densitometry of immunoblots (*n* = 3) from panel A. CD4 was normalized to β-actin as a loading control. Transfection efficiency was accounted for using FLAG abundance. (**C**) Representative images of immunoblot of pDNA-PKcs and total DNA-PKcs in HCT116 p300 WT and HCT116 p300 KO. (**D**) Densitometry of pDNA-PKcs normalized to total DNA-PKcs and GAPDH as a loading control. Data is shown relative to HCT116 WT control. (**E**) Representative images of pDNA-PKcs foci following Zeocin exposure. (**F**) Percentages of pDNA-PKcs foci positive (>2) HCT116 cells following Zeocin exposure. All microscopy images are 400X magnification. (**G**) Representative immunoblot of pArtemis and total Artemis of HCT116 cells. (**H**). Densitometry of immunoblots (*n* = 4) of pArtemis normalized to total Artemis, and to GAPDH as a loading control. All values are represented as mean ± standard error, from at least three independent experiments. Statistical differences between groups were measured by using Student’s *t*-test. * indicates *p* < 0.05. ** indicates *p* < 0.01. !! indicates significant difference between Zeocin treated and untreated group.

**Figure 7 cancers-12-02356-f007:**
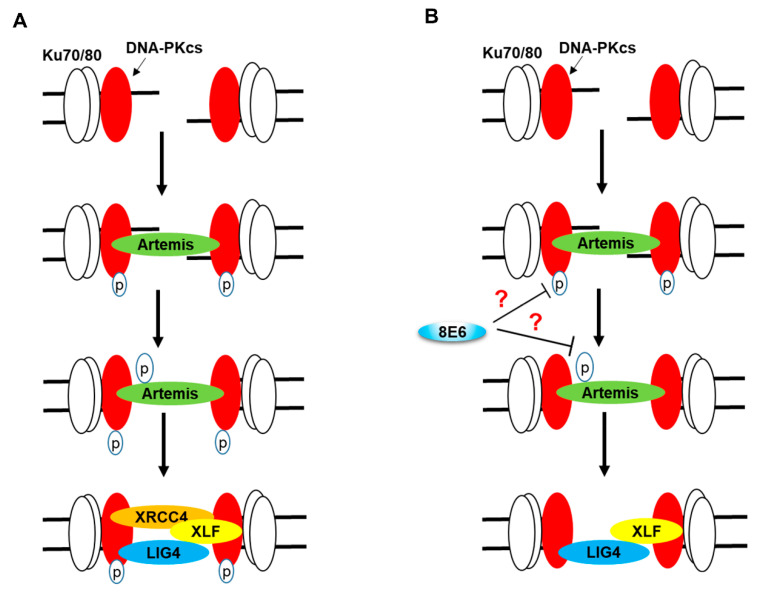
NHEJ in Cells with and without β-HPV 8E6. (**A**) Schematic of canonical NHEJ pathway. DNA-PK holoenzyme (Ku70/80/DNA-PKcs) binds to DSB, leading to DNA-PKcs autophosphorylation. Activated DNA-PK leads to Artemis phosphorylation and DNA end processing. Finally, the XRCC4/XLF/LIG complex repairs the break. (**B**) Schematic of β-HPV 8E6 alterations in canonical NHEJ. β-HPV 8E6 hinders DNA-PKcs autophosphorylation and activation, by which downstream steps, including Artemis phosphorylation and XRCC4 recruitment, were diminished. “?” represents unknown mechanism.

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
