# Peer review of "Beta Human Papillomavirus 8E6 Attenuates Non-Homologous End Joining by Hindering DNA-PKcs Activity"

_cancers, 2020, doi:10.3390/cancers12092356_

Round 1
Reviewer 1 Report
The authors describe the mechanism of the genome stabilizing of beta HPV8E6. We have already known that beta HPV infect cutaneous epithelia and may contribute towards the initiation of non-melanoma skin cancers. Its E6 proteins increase the steady-state levels of p53 in squamous epithelial cells. Overall the NEHJ story is very novel and intriguing. I have some concerns before accepting the article for publication.
Figure 1: The authors used U2OS cells to see NHEJ efficiency. It would be better to confirm the study using other cells such as HFK as a control.
Figure 2: C: Western blotting. I do not think the relative expression level of pDNA-PKcs downregulated in U2OS-β-HPV8E6 cells, as shown in Figure D densitometry. Why are the DNA-PKc levels decreasing by Zeocin treatment in a concentration-dependent manner in the delta 8E6 U2OS cells? Moreover, the result of the Zeocin effect in the delta 8E6 HFK cells is needed (Figure 2A) to compare the effect between the cells.
Figure3, 4, and 5: How about the results of pDNA-PKc foci in HFK delta HPVE6 cells?
Figure 5E: High-resolution figures are needed to see it in detail.
Figure 6: Why did the authors use HCT116 cells? It is difficult to conclude the story using these cells created from different organs. It would be better to use the same cells used in Figure 1-4.
Minor concern:
Abstract: It is better to describe the abbreviation of PK.
Author Response
Reviewer 1:
The authors describe the mechanism of the genome stabilizing of beta HPV8E6. We have already known that beta HPV infect cutaneous epithelia and may contribute towards the initiation of non-melanoma skin cancers. Its E6 proteins increase the steady-state levels of p53 in squamous epithelial cells. Overall the NEHJ story is very novel and intriguing. I have some concerns before accepting the article for publication.
Figure 1: The authors used U2OS cells to see NHEJ efficiency. It would be better to confirm the study using other cells such as HFK as a control.
Thank you for this suggestion. We measured NHEJ efficiency in HFKs (with and without 8E6) and include these data in our resubmission. Compared to U2OS cells, the transfection efficiency was notably reduced in these cells and particularly in HFKs expressing 8E6. Corrections based on transfection efficiency were included in the quantification shown in 1G. (Please see Figure 1F and G.) These results confirm HPV 8E6’s ability to decrease NHEJ efficiency in keratinocytes.
Figure 2: C: Western blotting. I do not think the relative expression level of pDNA-PKcs downregulated in U2OS-β-HPV8E6 cells, as shown in Figure D densitometry.
The induction in pDNA-PKcs was lower in the U2OS background. However, the reduced phosphorylation of DNA-PKcs was consistent across multiple (n=3) independent replicates. The differences are most notable in cells treated with the highest concentration of Zeocin (640 μg/ml).
Why are the DNA-PKc levels decreasing by Zeocin treatment in a concentration-dependent manner in the delta 8E6 U2OS cells?
Our interest was also drawn to these results. We believe that it may point to a mechanism by which Δ8E6 decreases NHEJ efficiency. However, we feel testing this hypothesis would be best addressed in a separate manuscript. In response to the reviewers comment, we now included a discussion of these ideas in our resubmission. (Please see lines 313-318)
Moreover, the result of the Zeocin effect in the delta 8E6 HFK cells is needed (Figure 2A) to compare the effect between the cells. Figure3, 4, and 5: How about the results of pDNA-PKc foci in HFK delta HPVE6 cells?
We appreciate this suggestion, but tend to agree with the second reviewer that our data are sufficient to justify our conclusions. Further, there are several reasons to believe additional confirmation that HPV 8E6’s attenuation of NHEJ does not notably differ between U2OS and HFK cells. Specifically, HPV 8E6 maintains an approximately equal ability to impair NHEJ activity in multiple assays regardless of cell type. This includes an NHEJ reporter assay, pDNA-PKcs foci intensity and persistence, induction of DNA-PKcs phophoryaltion, and the phosphorylation of Artemis. For these reasons, we have to politely disagree with the reviewer on this point with the exception of our XRCC4. In our original submission, we did not repeat this result in U2OS cells. We now provide these data. They show that HPV 8E6 diminishes zeocin-induced XRCC4 foci in U2OS cells. Once again confirming HPV 8E6 similarly dysregulates NHEJ signaling in HFK and U2OS cells. Interestingly, the attenuation of XRCC4 foci formation continued in HPV Δ8E6 expressing cells (Please see Figure 5G and 5H). The addition of this data also provides insight into the p300-independent mechanisms by which HPV 8E6 hinders NHEJ. We discuss this further in our response to reviewer 2’s third comment.
Figure 5E: High-resolution figures are needed to see it in detail.
In our resubmission, we provide high resolution figures. (Please see figure 5E.)
Figure 6: Why did the authors use HCT116 cells? It is difficult to conclude the story using these cells created from different organs. It would be better to use the same cells used in Figure 1-4.
The intention of figure 6 is to confirm p300’s role in NHEJ. The data shown reproduces the attenuation of NHEJ by HPV 8E6 mediated p300 destabilization. We feel that showing that p300 is required for robust NHEJ in this cell background is important for at least two reasons. First, it largely rules out concerns that the reported phenotypes are cell type specific. Second, it confirms specificity of the proposed mechanism of action, suggesting that the inability of β-HPV Δ8E6 to hinder DNA-PKcs signaling cannot be attributed to the mutation causing a complete loss of function).
Minor concern:
Abstract: It is better to describe the abbreviation of PK.
Thank you for spotting our short coming. This has been corrected in our resubmission. (Please see line 21.)

Reviewer 2 Report
While β-HPVs have been found to infect both immune compromised as well as immune competent individuals, infections are transient and a proposed role in non-melanoma skin cancer has been hard to document. The bulk of evidence to date support such a role, however recent evidence points towards a tumor protective role for these viruses in immune competent individuals (Strickley et.al. Nature 2019). This manuscript contributes towards understanding the ways in which β-HPVs may contribute to carcinogenesis by inhibiting the NHEJ pathway via its protein E6. This adds to a growing volume of evidence by this group and others which point to the conclusion that the E6 protein of β-HPVs may promote mutagenesis in infected cells by disabling multiple pathways implicated in DNA damage repair.
E6 was previously shown to impinge on the HR pathway. Using a previously described end-joining assay these authors showed that the efficiency of NHEJ in cells expressing β-HPV E6 was diminished. Pathway inhibition, as well as part of the mechanism through which this occurs was initially characterized in cell lines not relevant to HPV infection such as U2OS, and HCT116, but key points were verified in HFKs tranduced with E6-expressing retroviruses. The authors convincingly propose the reduction in NHEJ activity observed is likely linked to the observed diminished DNA-PKcs autophosphorylation and activation, and downstream steps including Artemis phosphorylation and XRCC4 recruitment. While part of the NHEJ inhibition mediated by E6 may be due to its association with p300, a mutant E6 which lacks the ability to bind p300 retains some activity, indicating that other interaction of E6 are also relevant.
The manuscript is interesting and well-written and the conclusions well-supported by the experimental evidence. However, some small improvements are required to make it suitable for publication:
- Regarding supplementary figure 2 it is best for the authors to provide quantification for their claim that β-HPV 8 E6 does not decrease NHEJ protein in untreated cells. In some cases it is not readily obvious from the blots.
- It would be useful for the authors to better discuss the ways in which p300 may be implicated in the control of NHEJ (not as clear as in the case of HR).
- Can the authors propose potential p300 independent ways in which E6 is regulating this pathway?
- How to the authors reconcile their findings with evidence that virus infection may have a protective effect against carcinogenesis?
Author Response
Reviewer 2:
While β-HPVs have been found to infect both immune compromised as well as immune competent individuals, infections are transient and a proposed role in non-melanoma skin cancer has been hard to document. The bulk of evidence to date support such a role, however recent evidence points towards a tumor protective role for these viruses in immune competent individuals (Strickley et.al. Nature 2019). This manuscript contributes towards understanding the ways in which β-HPVs may contribute to carcinogenesis by inhibiting the NHEJ pathway via its protein E6. This adds to a growing volume of evidence by this group and others which point to the conclusion that the E6 protein of β-HPVs may promote mutagenesis in infected cells by disabling multiple pathways implicated in DNA damage repair.
E6 was previously shown to impinge on the HR pathway. Using a previously described end-joining assay these authors showed that the efficiency of NHEJ in cells expressing β-HPV E6 was diminished. Pathway inhibition, as well as part of the mechanism through which this occurs was initially characterized in cell lines not relevant to HPV infection such as U2OS, and HCT116, but key points were verified in HFKs tranduced with E6-expressing retroviruses. The authors convincingly propose the reduction in NHEJ activity observed is likely linked to the observed diminished DNA-PKcs autophosphorylation and activation, and downstream steps including Artemis phosphorylation and XRCC4 recruitment. While part of the NHEJ inhibition mediated by E6 may be due to its association with p300, a mutant E6 which lacks the ability to bind p300 retains some activity, indicating that other interaction of E6 are also relevant.
The manuscript is interesting and well-written and the conclusions well-supported by the experimental evidence. However, some small improvements are required to make it suitable for publication:
1.Regarding supplementary figure 2 it is best for the authors to provide quantification for their claim that β-HPV 8 E6 does not decrease NHEJ protein in untreated cells. In some cases, it is not readily obvious from the blots.
We now provide densitometry for the NHEJ proteins shown in supplemental figure 2. (Please see supplemental figure 2B.)
2.It would be useful for the authors to better discuss the ways in which p300 may be implicated in the control of NHEJ (not as clear as in the case of HR).
We acknowledge that the link is not as clear as it was for HR, and now provide a plausible explanation of our results in our discussion. Specifically, a previous study demonstrated that p300 was required for Ku70/80 recruitment to DSB. These proteins along with DNA-PKcs form a holoenzyme that facilitates DNA-PKcs-mediated phosphorylation. (Please see lines 304-307.)
3.Can the authors propose potential p300 independent ways in which E6 is regulating this pathway?
We now include a discussion of possible ways that HPV 8E6 could be acting independently of p300. Specifically, this could be due to a decrease in the total DNA-PKcs abundance after induction of DSBs as shown in Figure 2C. Data added to this resubmission also demonstrate that HPV 8E6 inhibits XRCC4 foci formation in a p300-independent manner. (Please see lines 313-318 and Figure 5G and 5H.)
4.How to the authors reconcile their findings with evidence that virus infection may have a protective effect against carcinogenesis?
We do not see the ideas that β-HPV E6 is mutagenic and elicits a protective immune response as mutually exclusive. The oncogenic consequences (as the reviewer suggests) may be limited to immune compromised individuals. However, it should be noted that the Strickley et al relied almost exclusively on studies with MMuPV1 and the E6 protein from that virus does not bind p300. We have expanded our discussion to include these thoughts. (Please see lines 328-333.)

Round 2
Reviewer 1 Report
Thank you very much for this opportunity to review the manuscript. All of my concerns have been resolved by the authors' very informative messages.
Author Response
We appreciate your effort and valuable comments.